# Stress and burnout amongst mental health professionals in Singapore during Covid-19 endemicity

Suyi Yang[1]*, Germaine Ke Jia Tan[2], Kang Sim[3,4], Lucas Jun Hao Lim[4], Benjamin Yong Qiang Tan[5], Abhiram Kanneganti[6], Shirley Beng Suat Ooi[7,8], Lue Ping Ong[9]

1 Department of Occupational Therapy, Institute of Mental Health, Singapore, Singapore, 2 Department of Developmental Psychiatry, Institute of Mental Health, Singapore, Singapore, 3 West Region, Institute of Mental Health, Singapore, Singapore, 4 Department of Mood and Anxiety & West Region, Institute of Mental Health, Singapore, Singapore, 5 Department of Medicine, Yong Loo Lin School of Medicine, National University of Singapore, Singapore, Singapore, 6 Department of Obstetrics and Gynaecology, National University Hospital, Singapore, Singapore, 7 Emergency Medicine Department, National University Hospital, Singapore, Singapore, 8 Department of Surgery, Yong Loo Lin School of Medicine, National University of Singapore, Singapore, Singapore, 9 Allied Health Operations, Institute of Mental Health, Singapore, Singapore

* yang_suyi@imh.com.sg

**Data Availability Statement:** We have indicated that the data for this study is available upon request. The restriction is imposed by our institutional and ethics committee. Data can only be shared after a proposal is approved by the

## Abstract

The COVID-19 pandemic has exerted a huge emotional strain on mental health professionals (MHP) in Singapore. As Singapore transited into an endemic status, it is unclear whether the psychological strain has likewise lessened. The aims of this study were to investigate the levels of stress and burnout experienced by MHP working in a tertiary psychiatric hospital in Singapore during this phase of COVID-19 endemicity (2022) in comparison to the earlier pandemic years (2020 and 2021) and to identify factors which contribute to as well as ameliorate stress and burnout. A total of 282 MHP participated in an online survey in 2022, which included 2 validated measures, namely the Perceived Stress Scale and the Oldenburg Burnout Inventory (OLBI). Participants were also asked to rank factors that contributed the most to their stress and burnout. Between-group comparisons were conducted regarding stress and burnout levels among MHP across different demographic groupings and working contexts. In addition, OLBI data completed by MHP in 2020 and 2021 were extracted from 2 published studies, and trend analysis was conducted for the proportion of MHP meeting burnout threshold across 3 time points. We found that the proportion of MHP meeting burnout threshold in 2020, 2021 and 2022 were 76.9%, 87.6% and 77.9% respectively. Professional groups, age, years of experience and income groups were associated with stress and/or burnout. High clinical workload was ranked as the top factor that contributed to stress and burnout while flexible working arrangement was ranked as the top area for improvement so as to reduce stress and burnout. As such, policy makers and hospital management may want to focus on setting clear mental health targets and facilitate manageable clinical workload, build manpower resiliency, optimize resources and provide flexible work arrangements to alleviate stress and burnout among MHP.

ethics committee. This has also been conveyed to our participants during the consent process. The data request can be sent to The Institutional Research Review Committee, Institute of Mental Health, Singapore; Email address: imhresearch@imh.com.sg.

**Funding:** This research is supported by the Singapore Ministry of Health's National Medical Research Council under the Centre Grant Programme (Grant No.: NMRC/CG1/005/2021-IMH). The funder had no role in study design, data collection and analysis, decision to publish, or preparation of the manuscript.

**Competing interests:** The authors have declared that no competing interests exist.

## Introduction

Mental health professionals (MHP), in their daily interaction with patients, have to respond empathetically to the grief and anxiety expressed by their patients [1]. This in turn might lead to compassion fatigue and burnout [2]. Even before the COVID-19 pandemic began, there were concerns about the high levels of burnout experienced by MHP [3]. In a meta-analysis of thirty-three studies, O'Connor et al. [4] reported that the overall estimated pooled prevalence for emotional exhaustion and depersonalisation as measured by Maslach Burnout Inventory (MBI) among MHP was 40% and 22% respectively.

In the last few years, the emotional strain experienced by MHP was further aggravated by the COVID-19 pandemic [5]. To cope with a surge in demand of healthcare services while preventing further spread, hospitals in Singapore have had to implement various measures such as the split team arrangements, adherence to strict hygiene protocols and adoption of teleconsultation services. These rapidly changing guidelines, together with the personal fear of passing the virus to family members, have imposed an immense psychological burden on MHP. In a meta-analysis of 65 studies conducted during the pandemic on healthcare workers, Li et al. [6] reported that the estimated pooled prevalence for depression and anxiety was 21.7% and 22.1% respectively.

Singapore reported its first COVID-19 case in late January 2020 and by April 2020, strict lockdown measures were implemented for 2 months. During the 2 months, school and non-essential services were closed and work from home arrangements were encouraged for all essential services [5]. These restrictions were gradually lifted as conditions improved and effective vaccines were produced. By August 2021, the country has achieved a high vaccination rate of about 80%. From the beginning of 2022, Singapore began to transit into a state of COVID-19 endemicity and had started to approach travel and workplace normalcy by June 2022 [7]. However, it is unclear whether the stress and burnout levels have likewise lessened to an acceptable level. Thus, this study aims to (1) investigate the levels of stress and burnout experienced by MHP working in a tertiary psychiatric hospital in Singapore during this phase of COVID-19 endemicity (2022) in comparison to the earlier pandemic years (2020 and 2021), and (2) identify factors contributing to stress and burnout and areas of improvement to reduce stress and burnout using ANOVA.

## Methods

This study utilized a cross sectional survey study design. Waiver of consent and ethical approval was obtained from the National Healthcare Group Domain Specific Review Board in April 2022 (Reference number: 2021/01128). At the front page of the survey, prospective participants were informed that by submitting the survey, they were giving the study team implied consent to use the data collected for research.

### Data collection

An email with the survey link was sent to all MHP working at Institute of Mental Health (IMH) in Singapore in June 2022 via their work email address. The inclusion criteria included respondents (1) being MHP i.e. doctor, psychologist, social worker, occupational therapist, physiotherapist, pharmacist, nurse or case manager and (2) above the age of 21 years old. The responses were collected for a period of 1 month.

To analyse the trend in burnout among MHP from 2020 to 2022, we extracted a subset of the burnout scores completed by the MHP at the same hospital from 2 earlier multi-centre studies and compared them with the current study. As the 2 earlier studies did not collect stress data, only the burnout scores were extracted and compared. Both earlier studies utilized

an online survey to examine burnout among health care workers in Singapore and were conducted in May 2020 [5] and September 2021 [8].

## Measures

Demographic factors and work contexts were collated and stratified.

Perceived stress scale (PSS). The PSS is a self-reported questionnaire that measures the stress experienced by an individual over the past one month [9]. It has a total of 10 items scaled on Likert scales ranging from 0(never) to 4(very often). An item example is: "In the last month, how often have you been upset because of something that happened unexpectedly?" PSS is reported to have good internal consistency and factorial validity [9]. The Cronbach's alpha for the present study is 0.88.

Oldenburg burnout inventory (OLBI). The OLBI is a self-reported questionnaire that measures the levels of exhaustion and disengagement experienced by an individual [10]. Exhaustion refers to the experience of physical, cognitive or emotional strain while disengagement refers to distancing oneself from one's work [10]. It has a total of 16 items scaled on Likert scale ranging from 1(strongly agree) to 4(strongly disagree). An item example is: "There are days when I feel tired before I arrive at work." It has both negatively and positively worded questions, thus reducing response biases [11]. It is also reported to have good construct validity [11]. A mean score of $\geq$ 2.25 on the exhaustion subscale was determined as a cut-off threshold for meeting high exhaustion threshold, while a mean score of $\geq$ 2.1 on the disengagement subscale was determined as a cut-off threshold for meeting high disengagement threshold. A mean score of exhaustion $\geq$ 2.25 and disengagement $\geq$ 2.1 taken together were determined as cut-off for meeting burnout threshold [12]. The Cronbach's alpha for exhaustion and disengagement in the present study is 0.85 and 0.83 respectively.

Factors affecting stress and burnout and areas of improvement. Using unpublished qualitative data collected in an earlier burnout study [13], the authors have identified 9 factors that contributed to stress and burnout and seven areas for potential improvement (see Figs 1 and 2 for all the items). Participants were asked to rank the top three factors contributing to stress and burnout and the top three areas for improvement to reduce stress and burnout. In addition, participants were also asked to indicate whether they had sought help from the Staff Support and Assistance Program (SSAP) before and whether they found it useful. SSAP is an internal peer support program that provide psychological help to staff who experience work related stresses.

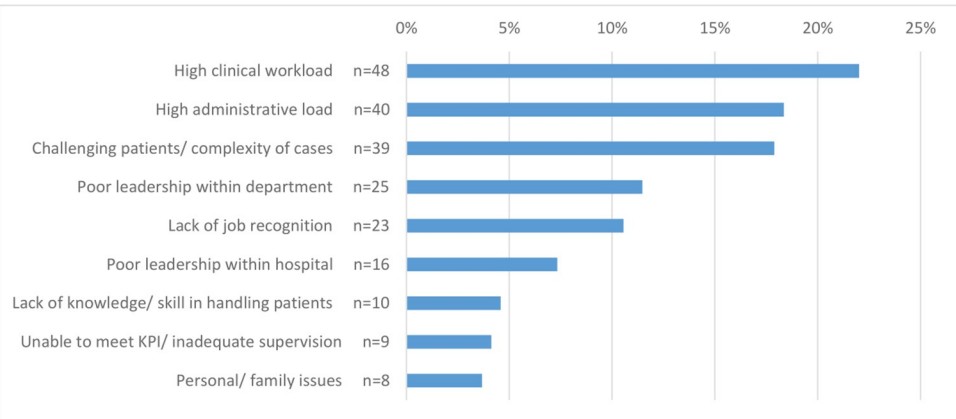

**Fig 1. Ranking of factors that most significantly contributed to stress and burnout in work settings.**

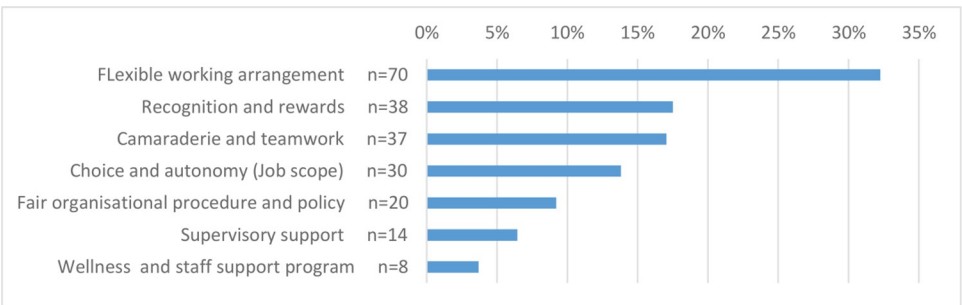

**Fig 2. Ranking of areas where potential improvements in the domains can most significantly reduce stress and burnout.**

## Data analysis

The data collected from the current study were analyzed using Jamovi Version 2.2.5. Descriptive statistics were calculated for all variables. Fisher ANOVA (equal variances) and Welch ANOVA (unequal variances) were used to examine any significant between-group differences in stress and burnout levels among MHP belonging to different demographic groupings and working situations. Post-hoc analysis using Tukey's (equal variances) and Games-Howell's tests (unequal variances), were conducted for variables that demonstrated a significant difference. P-value <0.05 was used to determine whether the ANOVA/ post-hoc tests were statistically significant.

## Results

### Survey responses

A total of 282 MHP participated in the survey conducted during June 2022 (see Table 1). However, as all the items are optional, there were missing data and total number of completed PSS and OLBI ratings was 239 and 231 respectively. The mean scores for PSS, OLBI-exhaustion and OLBI-disengagement were 19.7, 2.68 and 2.58 respectively.

### Comparisons across groups

One-way ANOVA was performed to compare the effect of demographic or work situation variables on level of stress, exhaustion and disengagement using the mean scores across groups. (1) Across MHP in different professional groups, level of exhaustion ($F_{(4,226)} = 2.9$, p = 0.02) and level of disengagement ($F_{(4,67)} = 3.14$, p = 0.02) differed significantly, (2) Across MHP with different years of experience, level of perceived stress ($F_{(3,235)} = 4.49$, p = 0.04), level of exhaustion ($F_{(3,227)} = 9.07$, p < .001) and level of disengagement ($F_{(3,227)} = 5.42$, p = 0.001) differed significantly, (3)Across MHP in different age groups, level of perceived stress ($F_{(3,227)} = 3.57$, p = 0.02), level of exhaustion ($F_{(3,226)} = 7.89$, p < .001) and level of disengagement ($F_{(3,226)} = 6.11$, p = 0.001) differed significantly, and (4) Across MHP in different income groups, level of exhaustion ($F_{(4,225)} = 3.38$, p = 0.01) differed significantly (see Fig 3 for descriptive plots and Table 2 for the post-hoc analyses). No other significant differences were found for other variables.

### Factors contributing to stress and burnout and areas for improvement

High clinical workload was ranked first by the greatest number of MHP when they ranked factors that contribute to their stress and burnout. Flexible working arrangement was ranked first

**Table 1. Summary of socio-demographic factors and work contexts (n = 282).**

| Variable | | n | % |
|---|---|---|---|
| Profession | Nurse | 64 | 22.7% |
| | Occupational therapist | 27 | 9.6% |
| | Dr/ Psychiatrist | 16 | 5.7% |
| | Social worker | 26 | 9.2% |
| | Case manager | 43 | 15.2% |
| | Psychologist | 20 | 7.1% |
| | Pharmacist | 17 | 6% |
| | Others | 69 | 24.5% |
| Age group | <30 | 57 | 24.7% |
| | 30–39 | 86 | 37.2% |
| | 40–49 | 53 | 22.9% |
| | >49 | 35 | 15.2% |
| Year of experience | <5 | 97 | 34.4% |
| | 6–10 | 61 | 21.6% |
| | 11–20 | 90 | 31.9% |
| | >20 | 34 | 12.1% |
| Income level | <S$30k | 23 | 10% |
| | S$30k-S$50k | 85 | 36.8% |
| | S$50k-S$80k | 73 | 31.6% |
| | S$80k-S$110k | 27 | 11.7% |
| | >S$110 | 23 | 10% |
| Employment type | Full-time | 271 | 96.1% |
| | Part-time/ locum | 11 | 3.9% |
| Work arrangement | Fully onsite | 231 | 81.9% |
| | Onsite/ home | 51 | 18.1% |
| Work area | Administrative | 63 | 22.4% |
| | Clinical | 210 | 74.7% |
| | Research | 8 | 2.8% |

by the greatest number of MHP when they ranked areas where improvement can help to reduce stress and burnout (see Figs 1 and 2).

## Staff support and assistance program

Only nine MHP reported that they have sought help from the SSAP, with three MHP who found it moderately or very helpful. Out of the 249 MHP that did not seek help, 111 MHP indicated that they did not need help, 66 MHP were concerned about confidentiality, while 72 MHP preferred to seek external help.

## Trend in burnout

Mean scores and proportions of MHP meeting the exhaustion, disengagement, and burnout thresholds across the 3 time points were presented in Table 3 and Fig 4.

## Discussion

This study offered insight into the level of burnout among MHP over three time points from 2020 to 2022. Our results indicated that the proportion of MHP meeting burnout threshold in May 2020, September 2021 and June 2022 were 76.9%, 87.6% and 77.9% respectively. When

Profession

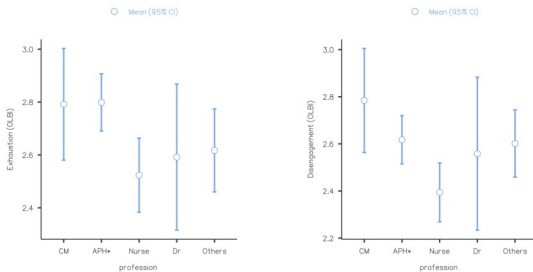

*Due to the small number in occupational therapists, psychologists, pharmacists and social workers categories, they were combined into one allied health professionals (AHP) group for between groups analysis.

Years of experiences

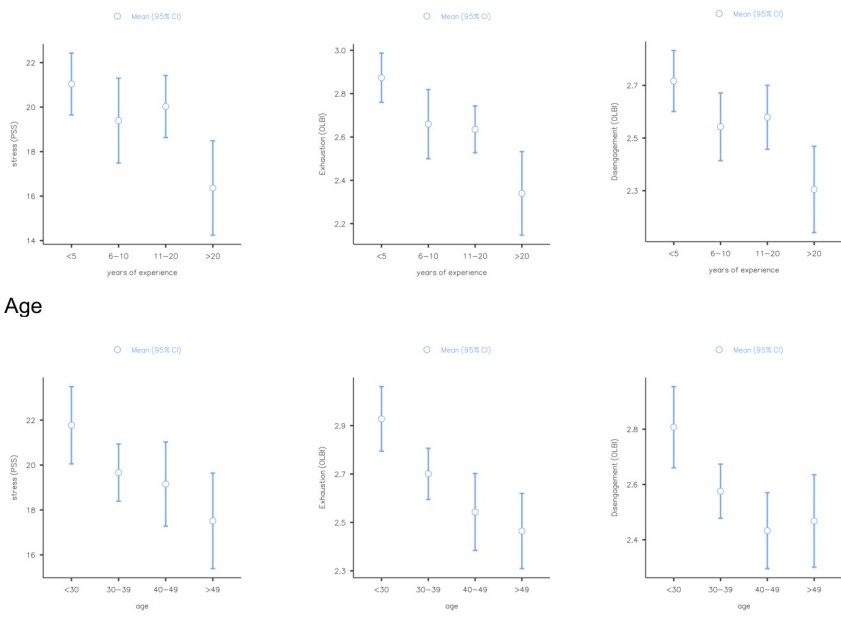

Age

Annual income

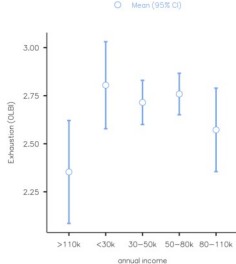

**Fig 3. Descriptive plots for stress, exhaustion and disengagement mean scores across different demographic or work situation variables.**

analysed together with the COVID-19 epidemic curve (see Fig 4), it appears that the increase in burnout rate in September 2021, when compared to May 2020, might be related to the increase in COVID-19 cases. Another possible reason for the increase in burnout rate could be due to high workload. During the implementation of strict lockdown measures in April and

**Table 2. Results of significant Tukey/ Games-Howell post-hoc tests for the one-way ANOVAs investigating relationships between demographic variables and both stress and burnout (exhaustion and disengagement).**

| **Dependent variable: Stress** | | | | | |
|---|---|---|---|---|---|
| Years of experience, N = 239 | | Mean difference | t value | df | p-value |
| <5 vs. | 6–10 | 1.64 | 1.43 | 235 | 0.48 |
| | 11–20 | 1.01 | 1.02 | 235 | 0.74 |
| | >20 | 4.67 | 3.63 | 235 | **0.002** |
| Age, N = 239 | | | | | |
| <30 vs. | 30–39 | 2.11 | 1.95 | 227 | 0.21 |
| | 40–49 | 2.62 | 2.17 | 227 | 0.13 |
| | >49 | 4.26 | 3.14 | 227 | **0.01** |
| **Dependent variable: Exhaustion** | | | | | |
| Professional groups, N = 231 | | Mean difference | t value | df | p-value |
| AHP vs. | Nurses | 0.28 | 2.99 | 226 | **0.03** |
| | Case managers | 0.007 | 0.07 | 226 | 1.0 |
| | Doctors | 0.21 | 1.41 | 226 | 0.62 |
| | Others | 0.18 | 1.91 | 226 | 0.31 |
| Years of experience, N = 231 | | | | | |
| <5 vs. | 6–10 | 0.21 | 2.27 | 227 | 0.11 |
| | 11–20 | 0.24 | 2.93 | 227 | **0.02** |
| | >20 | 0.53 | 5.07 | 227 | **< 0.001** |
| Age, N = 231 | | | | | |
| <30 vs. | 30–39 | 0.23 | 2.61 | 226 | 0.05 |
| | 40–49 | 0.38 | 3.94 | 226 | **< .001** |
| | >49 | 0.46 | 4.24 | 226 | **< .001** |
| Income Level, N = 216 | | | | | |
| >$110k vs. | <$30k | -0.45 | -2.94 | 225 | **0.03** |
| | $30k-$50k | -0.36 | -2.96 | 225 | **0.03** |
| | $50-$80k | -0.41 | -3.26 | 225 | **0.01** |
| | $80-$110k | -0.22 | -0.22 | 225 | 0.58 |
| **Dependent variable: Disengagement** | | | | | |
| Professional groups, N = 231 | | Mean difference | t value | df | p-value |
| Case managers vs. | AHP | 0.17 | 1.39 | 47.1 | 0.64 |
| | Nurses | 0.39 | 3.12 | 53.1 | **0.02** |
| | Doctors | 0.23 | 1.21 | 28.8 | 0.74 |
| | Others | 0.18 | 1.41 | 58.0 | 0.62 |
| Years of experience, N = 231 | | | | | |
| <5 vs. | 6–10 | 0.17 | 1.87 | 227 | 0.24 |
| | 11–20 | 0.14 | 1.72 | 227 | 0.32 |
| | >20 | 0.41 | 3.94 | 227 | **< .001** |
| Age, N = 231 | | | | | |
| <30 vs. | 30–39 | 0.23 | 2.73 | 226 | **0.03** |
| | 40–49 | 0.37 | 3.94 | 226 | **< .001** |
| | >49 | 0.34 | 3.19 | 226 | **0.009** |

May 2020, many non-emergency outpatient services were temporary halted. As services gradually resumed in the later part of 2020 and 2021, there were significant backlog of cases [5,8]. This, together with newly implemented infection control measures and increase in the number of people seeking mental health services because of the pandemic, resulted in an increase in workload in 2021 [8].

**Table 3. Measures of burnout across 3 studies.**

| | | Jun 2022 OLBI (n = 231) | | Sept 2021 OLBI (n = 169) | | May 2020 OLBI (n = 672) | |
|---|---|---|---|---|---|---|---|
| Scales | | Mean (SD) | Meeting threshold n(%) | Mean (SD) | Meeting threshold n(%) | Mean (SD) | Meeting threshold n(%) |
| OLBI_exhaustion | | 2.68(0.53) | 187(80.6%) | 2.73(0.45) | 153(90.5%) | 2.79(0.43) | 622(92.6%) |
| OLBI_disengagement | | 2.58(0.51) | 205(88.7%) | 2.66(0.46) | 155(91.7%) | 2.34(0.45) | 519(77.2%) |
| OLBI_meeting threshold for burnout* | | | 180(77.9%) | | 148(87.6%) | | 517(76.9%) |

*Burnout threshold: Exhaustion≥2.25 and disengagement≥2.1.

Interestingly, despite the surges of new COVID-19 cases in September 2021 and June 2022, there was a slight reduction in the burnout rate in June 2022. By June 2022, hospitals in Singapore have stepped down its COVID-19 infection control measures and most backlog of cases have being cleared. The reduced workload might have contributed to lower burnout rate. Another possible reason might be that as the country moved from COVID-19 pandemic to endemic status with high vaccination rates, there was a greater acceptance and less fear of contracting COVID-19. In addition, the lifting of several COVID-19 measures within all public hospitals such as lifting of restriction in staff gathering during this period might increase social engagement and reduce burnout among MHP.

Despite the slight reduction of burnout rate in June 2022, the proportion of MHP meeting threshold for burnout is still substantial. This is in line with existing literature reporting high burnout rates up to 71% amongst health care workers during the Covid-19 pandemic [14–16]. The only pre-pandemic study of burnout in MHP using OLBI in Singapore was done in 2014. Yang et al. [13], in their survey of 220 MHP, reported a mean exhaustion and disengagement scores of 2.48 and 2.34 respectively. This is lower than the mean of 2.68 and 2.58 reported in this current study. In another recent study conducted with 104 psychiatric residents in the national psychiatry residency program and who are rotated to sites across the nation including

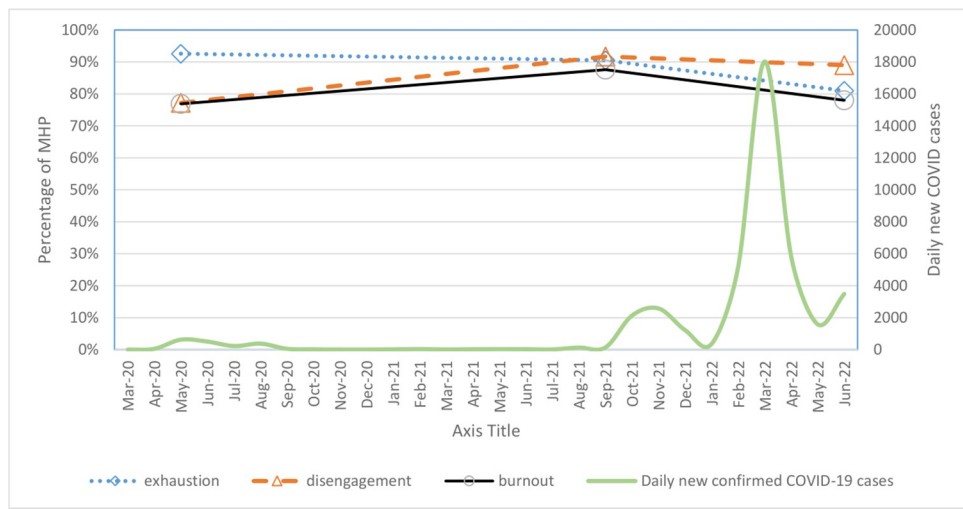

**Fig 4. Proportions of MHP meeting exhaustion, disengagement and burnout thresholds and daily new COVID cases.**

IMH, Chew et al. [17] reported that 54.8% of the respondents met the OLBI burnout threshold. Thus, it is likely that COVID-19 endemicity continues to have an impact on the well-being of MHP and that the burnout rate reported in this current study is higher than the pre-pandemic level.

When the proportion of MHP meeting the exhaustion and disengagement thresholds were analyzed across the 3 time points (see Table 3 and Fig 4), it appeared that the exhaustion rate (which peaked earlier at May 2020, maintained at Sept 2021 and decreased in June 2022) corresponded more immediately to COVID-19 related stresses in early 2020 while the disengagement rate (which peaked later at Sept 2021, then decreased slightly in June 2022) appeared to build up over a longer period in response to COVID-19 related stresses. This rise and fall trend is similar to that reported in a pandemic study conducted in Canada with 409 hospital workers across 4 time points [18]. These findings were in agreement with notion of Demerouti et al. [19] that individuals experience exhaustion faster than disengagement. In another study that focused on self-efficacy change as the mediator on the relationship between exhaustion and disengagement among 346 MHP, Rogala et al. [20] reported that exhaustion predicted disengagement about 6 months later and that self-efficacy beliefs have a mediating effect on the relationship. In addition to self-efficacy, other variables such as mindfulness and psychological flexibility can also mediate the relationship between exhaustion and disengagement [21,22]. More research is warranted in this area to help us better understand the inter-relationships between germane factors and identify strategies to reduce burnout.

When considering the level of stress experienced by MHP, the mean score of PSS-10 reported in this study (M = 19.7) was higher than that reported by Yang et al. [13] in their survey with MHP in Singapore (M = 18.7). When comparing with data from other countries conducted during the pandemic, our mean score was also higher. In a survey conducted with 227 healthcare professionals from 41 countries, the reported mean for PSS-10 was 17.6 [23]. In another study conducted in Columbia with 406 individuals, out of which 44% were health professionals, Pedrozo-Pupo et al. [24] reported that the mean score for PSS-10 was 16.5. Taken together with the OLBI data, our results suggested that MHP in Singapore were experiencing a high level of stress and burnout.

Findings from our June 2022 survey also revealed that the level of exhaustion and disengagement differed significantly across different professional groups. In particular, Allied Health Professions (AHP) (including occupational therapists, psychologists, pharmacists and social workers) reported the highest level of exhaustion and case managers reported the highest level of disengagement. One possible reason might be that during the start of the pandemic, AHP experienced a higher level of exhaustion as they were the main MHP to volunteer to man the National Care Hotline, which encompassed overnight shifts. The pandemic restrictions might also lead to a lower sense of self-efficacy as AHP were instructed to convert their in-person sessions into tele-consultations and needed to adjust to this mode of consultation. Case managers were also heavily taxed to check in on patients more frequently. Although AHP are no longer required to man the National Care Hotline and that most in-person sessions have resumed by June 2022, the cumulative effect of exhaustion experienced might still have an impact on their well-being.

Our findings also indicated that MHP age and years of experience were negatively associated with levels of stress, exhaustion and disengagement. This was consistent with findings of earlier studies whereby young and inexperienced MHP reported the highest level of burnout [4,18]. During the pandemic, many AHP leaders were concerned about inadequate in-person supervision for junior MHP. This was compounded by national-wide social distancing measures which further reduced social support. Several studies had shown that peer support was important to support MHP in their work, which can be emotionally demanding [2,25]. On the

other hand, the more experienced MHP might have already cultivated strong support networks and acculturated to the dynamic nature of healthcare demands, and thus were able to cope better. In addition, when compared to senior MHP, the junior staff may have varying clinical caseload and reduced autonomy in their job scope, thus contributing to burnout.

In our current study, high clinical workload was ranked first by the greatest number of respondents when they ranked factors that contributed to their stress and burnout. This is consistent with findings reported by a multi-centre prospective study conducted with healthcare workers in Singapore between March to August 2020 [26]. In their survey with 2744 healthcare workers working in 4 tertiary hospitals during COVID-19 pandemic, Teo el al. [26] reported that working long hours were strongly associated with stress and burnout.

In Singapore, there are about 4.5 psychiatrists and 8.9 psychologists per 100k population [27]. This is much lower as compared to developed countries like USA and Australia, where the psychiatrist to population ratio ranges from 11–18.3/100k and psychologist to population ratio ranges from 30–202.7/100k [28–30]. Some studies have suggested that workforce shortage affects MHP morale as they felt they were too busy to provide optimal care [4,31] and the gap between what MHP aspire to provide in clinical care and what is actually being delivered potentially contributes to burnout [32]. Even before COVID-19 pandemic, there have already been concerns about MHP shortage worldwide and the need to train more MHP [33–35]. The pandemic has indeed further accelerated this need [31].

Findings from this study also revealed that flexible working arrangement was ranked first by the greatest number of MHP when they ranked areas where improvements can help to reduce stress and burnout. It is likely that provision of flexible work arrangements can potentially increase the autonomy of MHP and help relieve family duties or facilitate balance between work and rest [36]. Of note, only a small proportion of MHP ranked improvement in supervisory support as most important. Most studies on burnout have advocated for more supervision for MHP [17,37]. However, it seemed that such support might be adequate for our respondents. Our study had also revealed that the utilization rate of SSAP was very low. To further improve staff wellness, a new recent arrangement has been initiated internally whereby MHP can seek external psychological help and further investigation into its effectiveness is needed in the future.

The relatively high rate of burnout highlighted in this study is consistent with the current literature and of concern as burnout is often associated with reduced therapeutic effectiveness [38], decreased job satisfaction [39], high turnover [40]. It is also considered economically wasteful given the high cost in training of MHP. Han et al. [41], in their study examining the cost of physician burnout in the US, estimated that physician burnout costs the country USD $4.6 billion each year due to higher staff turnover and reduced clinical hours.

Our study has highlighted the need for the development and implementation of health policies to address the burnout in MHP [42]. Healthcare policy makers and hospital management may want to focus on setting clear mental health targets and standardising measurement tools to assess and monitor burnout in MHP. There is also a pressing need to increase manpower capacity, improve workload sustainability and ensure that MHP skills and energies are not excessively expended. Optimising work processes and adoption of digital technology can be implemented to reduce administrative workload. Lastly, provision of flexible working arrangement can help to facilitate better work-life balance and improve health and well-being of MHP.

## Limitations

Our current study has a low response rate of 14.1%. There might also be a sampling bias as MHP who were experiencing burnout might not have participated in the survey. Moreover,

data were collected only from MHP working in one tertiary psychiatric hospital. Thus, the findings from this study cannot be generalized to all MHP working in Singapore. In addition, as this is a cross-sectional study, interpretation of causality is limited and we tried to mitigate this by comparing the burnout rate with the findings from 2 earlier local studies involving the same institution. The trending of burnout rate across 3 time points provides useful insight on burnout rate among MHP from 2020 to 2022. However, it is important to note that the burnout data from the 3 time points were drawn from different sample cohorts and a longitudinal study in which participants are follow-up using a prospective cohort study design would provide better insights into how burnout rate changes over time among MHP in Singapore.

## Conclusion

In conclusion, we found a relatively high level of Covid-19 related burnout amongst MHP which is in agreement with extant literature amongst healthcare workers. Even though we have transited into COVID endemicity at the time of the survey, the burnout rate in MHP has only reduced by a small proportion. Policy makers and hospital management may want to focus on setting clear mental health targets and strategies including proffering manageable clinical workload, building manpower capacity and resiliency, optimizing resources using digital technology, and the provision of flexible work arrangements to enhance the overall well-being of MHP.

## Acknowledgments

The authors would like to thank all subjects for their participation, Ms Chris N and IMH Research Division for their assistance in this study.

## Author Contributions

**Conceptualization:** Suyi Yang, Germaine Ke Jia Tan, Kang Sim, Lue Ping Ong.

**Data curation:** Suyi Yang, Germaine Ke Jia Tan, Lucas Jun Hao Lim, Benjamin Yong Qiang Tan, Abhiram Kanneganti.

**Formal analysis:** Suyi Yang, Germaine Ke Jia Tan.

**Funding acquisition:** Suyi Yang, Germaine Ke Jia Tan.

**Investigation:** Suyi Yang, Germaine Ke Jia Tan, Kang Sim, Lucas Jun Hao Lim, Benjamin Yong Qiang Tan, Abhiram Kanneganti, Shirley Beng Suat Ooi.

**Methodology:** Suyi Yang, Germaine Ke Jia Tan.

**Project administration:** Suyi Yang, Germaine Ke Jia Tan.

**Resources:** Suyi Yang, Germaine Ke Jia Tan.

**Software:** Suyi Yang, Germaine Ke Jia Tan.

**Supervision:** Kang Sim, Lue Ping Ong.

**Validation:** Suyi Yang, Germaine Ke Jia Tan.

**Visualization:** Suyi Yang, Germaine Ke Jia Tan.

**Writing – original draft:** Suyi Yang, Germaine Ke Jia Tan, Lue Ping Ong.

**Writing – review & editing:** Suyi Yang, Germaine Ke Jia Tan, Kang Sim, Lucas Jun Hao Lim, Benjamin Yong Qiang Tan, Abhiram Kanneganti, Shirley Beng Suat Ooi, Lue Ping Ong.

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
