## [Decision Letter · Decision Letter 0]

25 Apr 2023

PONE-D-22-33528Stress and burnout amongst mental health professionals in Singapore during Covid-19 endemicityPLOS ONE

Dear Dr. Yang Suyi

Thank you for submitting your manuscript to PLOS ONE. After careful consideration, we feel that it has merit but does not fully meet PLOS ONE’s publication criteria as it currently stands. Therefore, we invite you to submit a revised version of the manuscript that addresses the points raised during the review process.

We look forward to receiving your revised manuscript.

Kind regards,

Jasna Karacic Zanetti, Ph.D

Academic Editor

PLOS ONE

“This research is supported by the Singapore Ministry of Health’s National Medical Research Council under the Centre Grant Programme (Grant No.: NMRC/CG1/005/2021-IMH).”

“This research is supported by the Singapore Ministry of Health’s National Medical Research Council under the Centre Grant Programme (Grant No.: NMRC/CG1/005/2021-IMH). The funder had no role in study design, data collection and analysis, decision to publish, or preparation of the manuscript.”

Additional Editor Comments:

Dear authors,

it was my pleasure to read your article.

Please correct step by step all comments from the reviewers, as well you can mention how political decisions that are made affect burnout ( https://pubmed.ncbi.nlm.nih.gov/34828596/)

Reviewers' comments:

Reviewer's Responses to Questions

**Comments to the Author**

1. Is the manuscript technically sound, and do the data support the conclusions?

Reviewer #1: Yes

Reviewer #2: Partly

2. Has the statistical analysis been performed appropriately and rigorously? 

Reviewer #1: No

Reviewer #2: No

3. Have the authors made all data underlying the findings in their manuscript fully available?

Reviewer #1: Yes

Reviewer #2: No

4. Is the manuscript presented in an intelligible fashion and written in standard English?

Reviewer #1: Yes

Reviewer #2: Yes

5. Review Comments to the Author

Reviewer #1: Thank you for allowing me to review this interesting paper on stress and burnout amongst mental health professionals in Singapore during COVID-19 endemicity. As COVID-19 is part of our everyday lives currently, and as mental health issues gain prominence, making sure that mental health professionals who provide care to patients that need this help is very important. This paper has the potential to inform clinical practice.

I have a few minor suggestions for the authors to consider:

1. In the statistical section, state what p value as considered significant.

2. It would be helpful to provide some context on the clinical workload of the centre, including how this changed during the different time periods, even if discussed in a qualitative fashion that would be helpful.

3. Discuss the potential impact of the limitations on the estimates seen in the results.

Reviewer #2: please refer to my reviewers comments that i have attached above.

I think this it is an important topic.

The study aimed to highlight the high prevalence of burn out and stress in the mental health professionals

Nonetheless the low response rate of the survey limits the strength of the study to draw meaningful conclusions.

The comparative data was drawn from previous publications without elaborations of consistency of methodology or study population

Authors should also consider drawng references from previous multi-institutional, longitudinal study that were conducted in similar healthcare setting.

kindly refer to my atached comment above.

6. PLOS authors have the option to publish the peer review history of their article (what does this mean?). If published, this will include your full peer review and any attached files.

Reviewer #1: No

Reviewer #2: No

---

## [Author Response · Author response to Decision Letter 0]

28 Apr 2023

(Please refer to the attached "Responses to reviewers" file.)

Dear Editor and reviewers,

Thank you for your feedback and suggestions. Please find below the responses to each point raised by the editor/ reviewers.

Responses to editor 

Please correct step by step all comments from the reviewers, as well you can mention how political decisions that are made affect burnout. 

Response: Thank you for highlighting the importance of health politics. We have added more suggestions on how health policy can improve the well-being of MHP in p19 (Discussion section).

Responses to reviewer#1

Reviewer #1: Thank you for allowing me to review this interesting paper on stress and burnout amongst mental health professionals in Singapore during COVID-19 endemicity. As COVID-19 is part of our everyday lives currently, and as mental health issues gain prominence, making sure that mental health professionals who provide care to patients that need this help is very important. This paper has the potential to inform clinical practice.

I have a few minor suggestions for the authors to consider:

1. In the statistical section, state what p value as considered significant.

Response: Thank you for the feedback and suggestions. We have added in “P-value <0.05 was used to determine whether the ANOVA/ post-hoc tests were statistically significant” in p7 (Data analysis section). 

2. It would be helpful to provide some context on the clinical workload of the centre, including how this changed during the different time periods, even if discussed in a qualitative fashion that would be helpful.

Response: Thank you for the suggestion. We have added in some context on the clinical workload of MHP and how it changed during the different time periods in p13-14 (Discussion section).

3. Discuss the potential impact of the limitations on the estimates seen in the results.

Response: We are unsure what estimates you are referring to here. We have indicated in the limitation section that due to low response rate, finding cannot be generalised to all MPH working in Singapore). We have also added that use of different sample cohort limits interpretation of causality and a longitudinal follow-up study is recommended. These can be found in p 19 (Limitation section). 

Responses to reviewer#2

Reviewer #2: please refer to my reviewers comments that i have attached above.

I think this it is an important topic. The study aimed to highlight the high prevalence of burn out and stress in the mental health professionals. Nonetheless the low response rate of the survey limits the strength of the study to draw meaningful conclusions. The comparative data was drawn from previous publications without elaborations of consistency of methodology or study population

Authors should also consider drawing references from previous multi-institutional, longitudinal study that were conducted in similar healthcare setting.

Response to item 1: Thank you for your feedback and suggestions. The first study in May 2020 was carried out 2 months after strict lockdown measures were implemented in April 2020 in Singapore. The aim was to collect baseline burnout data among healthcare workers. The second study in September 2021 was carried out after about 80% of the population have been vaccinated against COVID-19 by August 2021. Despite the high vaccination rate in Singapore, the healthcare system was still strained by a high volume of COVID-19 cases and it was unclear whether there was any change in burnout rate among healthcare workers. The aim of the September 2021 study was to compare the burnout rate to the baseline rate. The third study in June 2022 was carried out 6 months after Singapore has transited into a state of COVID-19 endemicity. The aim was to assess whether burnout among healthcare workers has lessened to an acceptable level during COVID-19 endemicity. We have added a summarized timeline of the major covid related events in p.3-4 (Introduction section). This will help provide context to the timing of the 3 studies.

Response to item 2: The decision to implement the study was based on major COVID-19 related events as explained in point 1. We did not time our studies based on when the COVID-19 waves spikes as these spikes are hard to predict. We have added how some COVID-19 related events might have an impact on MHP burnout rate in p.13-14 (Discussion section)

Response to item 3: The possible factors that may aggravate or mitigate stress and burnout were drawn from a study conducted in the same psychiatry hospital (IMH) in 2014 by the same author. Though only the quantitative findings were published then, the qualitative findings were analyzed internally to provide a better understanding of causes of burnout among mental health professionals. These findings were contributed by mental health professionals working in the same workplace (with similar work conditions and work culture) and provide valuable and relevant insights into possible factors that may aggravate or mitigate stress and burnout for mental health professionals working in IMH. Thus they were used to formulate the survey question in this study. 

In the discussion section, we have referenced some studies from Asian region when discussing factors that impact stress and burnout. Please refer to page 16-17. Thank you for bringing our attention to Teo el al study. We have added a discussion point in reference to this study in p16 (Discussion section).

Response to item 4: Yes, we agree that the low response rate and the use of different sample cohorts to trend the burnout rate is a limitation to the study. We have added these limitations in p 19 (Limitation section).

---

## [Decision Letter · Decision Letter 1]

11 Sep 2023

PONE-D-22-33528R1Stress and burnout amongst mental health professionals in Singapore during Covid-19 endemicityPLOS ONE

Dear Dr. Suyi,

Thank you for submitting your manuscript to PLOS ONE. After careful consideration, we feel that it has merit but does not fully meet PLOS ONE’s publication criteria as it currently stands. Therefore, we invite you to submit a revised version of the manuscript that addresses the points raised during the review process.

We look forward to receiving your revised manuscript.

Kind regards,

Jasna Karacic Zanett

Academic Editor

PLOS ONE

Journal Requirements:

**Additional Editor Comments:**

Dear Authors,

Thank you for your updated version of the manuscript. Please consider the comments from the review and then send it back to us again.

Reviewers' comments:

Reviewer's Responses to Questions

**Comments to the Author**

1. If the authors have adequately addressed your comments raised in a previous round of review and you feel that this manuscript is now acceptable for publication, you may indicate that here to bypass the “Comments to the Author” section, enter your conflict of interest statement in the “Confidential to Editor” section, and submit your "Accept" recommendation.

Reviewer #3: All comments have been addressed

Reviewer #4: (No Response)

2. Is the manuscript technically sound, and do the data support the conclusions?

Reviewer #3: Yes

Reviewer #4: Yes

3. Has the statistical analysis been performed appropriately and rigorously? 

Reviewer #3: Yes

Reviewer #4: Yes

4. Have the authors made all data underlying the findings in their manuscript fully available?

Reviewer #3: Yes

Reviewer #4: Yes

5. Is the manuscript presented in an intelligible fashion and written in standard English?

Reviewer #3: Yes

Reviewer #4: Yes

6. Review Comments to the Author

Reviewer #3: Please go over the manuscript once again to avoid any minor grammatical, punctuation and consistent style of writing.

Thank you for adequately revising.

Reviewer #4: Overall, there are grammatically some incorrect parts (Line 369:exitant,line 91:this 2 months, line 266: exitant, line108:April2022, line 112 :June2022)->grammatical correction required

Methodology

-The author should show a sampling method used (how participants have been chosen from population) and the period of data collection

-it would be good to add Ethical part that would gather clear information on ethic (from line 106-108 and Line 115-117). Some informations of ethic appear in the study methods and data collection.

-I would like to know if the data extracted from burnout scores from two early multi-centre studies were only about burnout, as there is no information on stress from the two studies?

-It would good to state the statistical method used on each specific objective, as there is information on the statistical method used to assess factors contributing to stress and burnout and areas of improvement.

-There is a need of explanation in methodology part on how analysis has been made on each items(stress and burnout) as it seems the three time comparison has been made on burnout only while for stress the information is given on the survey conducted by the author in this study(year 2022)

Discussion

In reducing the discussion part Somme parts should be removed. For example the information given in lines 350-353 and lines 364-366 should be removed or moved at the last paragraph.

7. PLOS authors have the option to publish the peer review history of their article (what does this mean?). If published, this will include your full peer review and any attached files.

Reviewer #3: No

Reviewer #4: **Yes: **Innocent Yandemye

---

## [Author Response · Author response to Decision Letter 1]

5 Oct 2023

Dear Editor and reviewers,

Thank you for your feedback and suggestions. Please find below the responses to each point raised by the reviewers.

Responses to reviewer#4

Overall, there are grammatically some incorrect parts (Line 369:exitant,line 91:this 2 months, line 266: exitant, line108:April2022, line 112 :June2022)->grammatical correction required

The above listed grammatical errors have been corrected. 

Methodology

-The author should show a sampling method used (how participants have been chosen from population) and the period of data collection

All staffs working at IMH were send the email with the survey link and the period of data collection was one month. These are now added in the methodology (data collection) section.

-it would be good to add Ethical part that would gather clear information on ethic (from line 106-108 and Line 115-117). Some informations of ethic appear in the study methods and data collection.

Line 115-117 have been moved to the ethical section (starting with line 106) as suggested.

-I would like to know if the data extracted from burnout scores from two early multi-centre studies were only about burnout, as there is no information on stress from the two studies?

Yes, the earlier 2 studies did not collect stress (PSS) data.

-It would good to state the statistical method used on each specific objective, as there is information on the statistical method used to assess factors contributing to stress and burnout and areas of improvement.

We have added in line103 that the method used to assess factors contributing to stress and burnout is ANOVA in line 106.

-There is a need of explanation in methodology part on how analysis has been made on each items(stress and burnout) as it seems the three time comparison has been made on burnout only while for stress the information is given on the survey conducted by the author in this study(year 2022)

As the earlier 2 studies did not collect stress (PSS) data, only burnout (OLBI) data is available for comparison across 3 timepoint. We have added this in line 125-127.

Discussion

In reducing the discussion part Somme parts should be removed. For example the information given in lines 350-353 and lines 364-366 should be removed or moved at the last paragraph.

Line 350-353 have been removed as suggested. However, we have kept line 364-366 (line 371-373 in the revised manuscript with track change) as it is useful information to know that an external help service is now available and that further investigation into its effectiveness is needed in the future. This information fits better in its current place than in the last paragraph.

---

## [Decision Letter · Decision Letter 2]

19 Dec 2023

Stress and burnout amongst mental health professionals in Singapore during Covid-19 endemicity

PONE-D-22-33528R2

Dear Dr. Suyi,

We’re pleased to inform you that your manuscript has been judged scientifically suitable for publication and will be formally accepted for publication once it meets all outstanding technical requirements.

Kind regards,

Kenji Hashimoto, PhD

Section Editor

PLOS ONE

Additional Editor Comments (optional):

Reviewers' comments:

Reviewer's Responses to Questions

**Comments to the Author**

1. If the authors have adequately addressed your comments raised in a previous round of review and you feel that this manuscript is now acceptable for publication, you may indicate that here to bypass the “Comments to the Author” section, enter your conflict of interest statement in the “Confidential to Editor” section, and submit your "Accept" recommendation.

Reviewer #4: (No Response)

2. Is the manuscript technically sound, and do the data support the conclusions?

Reviewer #4: Yes

3. Has the statistical analysis been performed appropriately and rigorously? 

Reviewer #4: Yes

4. Have the authors made all data underlying the findings in their manuscript fully available?

Reviewer #4: Yes

5. Is the manuscript presented in an intelligible fashion and written in standard English?

Reviewer #4: Yes

6. Review Comments to the Author

Reviewer #4: The author should remove test method from the end of Introduction part. I was stisfied with the response of test method, there is no need to add test methodology in Introduction part

7. PLOS authors have the option to publish the peer review history of their article (what does this mean?). If published, this will include your full peer review and any attached files.

Reviewer #4: No

---

## [Editor Report · Acceptance letter]

4 Jan 2024

PONE-D-22-33528R2 

PLOS ONE

Dear Dr. Yang, 

I'm pleased to inform you that your manuscript has been deemed suitable for publication in PLOS ONE. Congratulations! Your manuscript is now being handed over to our production team.

Kind regards, 

on behalf of

Prof. Kenji Hashimoto 

Section Editor

PLOS ONE